# 3D Autoencoding Diffusion Model for molecule interpolation and manipulation

## Abstract

Manipulating known molecules and interpolating between them is useful for many applications in drug design and protein engineering, where exploration around known molecule(s) is involved. Recent studies using equivariant diffusion models have made significant progress in the *de novo* generation of high-quality molecules, but using these models to directly manipulate a known 3D molecule or a pair of 3D molecules remains less explored. This is mainly due to an intrinsic property of diffusion models: the lack of a latent semantic space that is easy to operate on. To address this issue, we propose the first semantics-guided equivariant diffusion model that leverages the "semantic" embedding of a 3D molecule, learned from an auxiliary encoder, to control the generative denoising process. Consequently, by modifying the embedding, we can steer the generation towards another specified 3D molecule or a desired molecular property. We show that our model can effectively manipulate basic chemical properties, outperforming several baselines. We further verify that our approach can achieve smoother interpolation between 3D molecule pairs compared to standard diffusion models.

## 1 Introduction

The 3D geometry of molecules holds significant implications for their properties and functions, such as quantum chemical properties (Fuchs et al., 2020; Finzi et al., 2020), molecular dynamics (Doerr et al., 2020; Wu & Li, 2023; Musaelian et al., 2023), and interactions with protein receptors (Anderson, 2003; Lim et al., 2019). Several deep learning techniques have been successfully applied to the generation of 3D molecules, including autoregressive generative models (Gebauer et al., 2019; Masuda et al., 2020; Luo et al., 2021) and equivariant normalizing flows (Satorras et al., 2021a).

More recently, diffusion models (Ho et al., 2020; Song et al., 2020) have achieved state-of-the-art results in generating high-quality 3D molecules (Hoogeboom et al., 2022; Xu et al., 2023). Diffusion models belong to the class of generative models designed to approximate the data distribution by iteratively removing noise. Unlike previous methods that rely on autoregressive generation, diffusion models perform simultaneous refinement of all elements, such as atoms, in each denoising iteration. This collective refinement approach enables them to effectively capture the underlying data structures, making them well-suited for structured generation such as molecule generation. Several recent studies further incorporate auxiliary information to improve the quality of generated molecules (Huang et al., 2023; Vignac et al., 2023).

Most of these existing works, however, focus on *de novo* generation, i.e. generating random 3D molecules. This has many applications such as drug design (Jin et al., 2018; Zang & Wang, 2020) and protein engineering (Lippow et al., 2007; Jin et al., 2021). In these applications, we want to maintain the backbone and certain properties of a known molecule with minimal modifications while modifying target properties.

To attain the aforementioned goal, we propose 3D autoencoding diffusion model for molecule interpolation and manipulation in this paper. Inspired by the idea of high-level semantic guidance of diffusion models for controllable generation of images, we learn the semantic embedding of a 3D molecule by an auxiliary encoder and use it as a condition for an equivariant diffusion model, resulting in an autoencoder-like architecture for 3D molecule generation. Consequently, the semantic embedding determines most of the compositional and structural information of a 3D molecule. We

could then make use of the semantics embedding to generate molecules with both desired composition and property. Specifically, we can directly operate on the embeddings for interpolation and property manipulation. We call our method `egDiffAE`. To our knowledge, this is the first approach to achieve 3D semantics-guided molecule manipulation with diffusion models.

We summarize the main contributions of our work as follows:

- We devise a semantics-guided autoencoding diffusion model for the generation of 3D molecules, where the semantic embedding controls the majority of the generated molecules.
- We show that with the semantic embedding, our method could achieve smoother interpolation between known molecules compared to standard diffusion models.
- We report the highest property manipulation performance in both error towards the target value as well as generation quality over several baselines.

## 2 RELATED WORK

**Controllable Generation with Diffusion Models** Controllable generation on a typical diffusion model is challenging as it lacks an explicit latent space to operate on. The latent diffusion model (Rombach et al., 2022) is trained on the low-dimensional latent space of another generative model, but sampling from that latent space is still difficult to control. Thus, controllable generation is usually achieved by external guidance on the denoising process. For example, classifier guidance (Dhariwal & Nichol, 2021) and classifier-free guidance (Ho & Salimans, 2022) could generate random samples conditioned on class labels or continuous properties. Recent studies further attempt to control the semantics of the generated samples, such as the content of images or the general look of faces. Kwon & Ye (2022) enforce a similarity score with the source image during the generation. Preechakul et al. (2022) use the semantics embedding to guide the denoising process. Wang et al. (2023) further regularizes the distribution of the embedding to facilitate random sampling. These methods are designed for image or video generation, while our proposed method leverages the semantic guidance with further regularization to control the properties in 3D molecule generation.

***De novo* Molecular Generation in 3D** Recently, the generation of 3D molecules has been an active area of research. To properly represent the 3D geometry, some methods learn a joint distribution between several geometric features (Luo et al., 2022; Anand & Achim, 2022). Others directly operate on the 3D molecular space. G-SchNet autoregressive samples 3D molecules with symmetry constraints (Gebauer et al., 2019). More recently, equivariant diffusion models (Hoogeboom et al., 2022; Xu et al., 2022; 2023) show promising results in generating high-quality molecules, and further in the generation tasks with external conditioning, such as designs for ligands (Schneuing et al., 2022; Torge et al., 2023) and linkers (Igashov et al., 2022). However, most of them are *de novo* with no explicit control over the content of the generated molecules. For our application, we would like to modify certain aspects of the generated samples (e.g. a chemical property) while maintaining the rest (e.g. the overall shape and composition), which requires controllable generation based on higher-level semantics.

**Molecular Manipulation in 2D** Beyond fully *de novo* generation, several applications modify known molecules, such as lead compound optimization in drug designs (Jorgensen, 2009). In those cases, the generated molecule should be sufficiently similar to the template, while some properties are modified. Maziarka et al. (2020) utilize a CycleGAN (Zhu et al., 2017) framework to map the template molecule from one domain (without the property) to another (with the property). Other works aim at manipulating a lower-dimensional latent space (Gómez-Bombarelli et al., 2018; Jin et al., 2018; Zang & Wang, 2020; Du et al., 2022; Wang et al., 2022). The majority of the existing methods rely on 2D graphs with autoregressive decoding. To our knowledge, very few study the challenging problem of controllable generation of 3D molecules, which is the aim of this paper.

## 3 BACKGROUND

### 3.1 DIFFUSION MODELS

The denoising diffusion probabilistic model (DDPM) (Ho et al., 2020) approximates the data distribution $p_\theta(\boldsymbol{x}_0)$ through a series of latent variables $\boldsymbol{x}_1, ..., \boldsymbol{x}_T$ from the same space as the data

$\boldsymbol{x}_0$, starting from a random noise point $\boldsymbol{x}_T$:

$$p_\theta(\boldsymbol{x}_0) = \int p(\boldsymbol{x}_T) \prod_{t=1}^{T} p_\theta(\boldsymbol{x}_{t-1}|\boldsymbol{x}_t)\mathrm{d}\boldsymbol{x}_{1:T},$$

This can be considered as progressively removing the noise from corrupted data $\boldsymbol{x}_T$, which is called the *reverse* or *denoising process*. The posterior $q(\boldsymbol{x}_{1:T}|\boldsymbol{x}_0)$, or the *forward* or *noising process*, is a process of gradually adding noise to $\boldsymbol{x}_0$ until it eventually becomes a random noise $\boldsymbol{x}_T$:

$$q(\boldsymbol{x}_{1:T}|\boldsymbol{x}_0) = \prod_{t=1}^{T} q(\boldsymbol{x}_t|\boldsymbol{x}_{t-1}).$$

The objective is to maximize the ELBO of $\log p(\boldsymbol{x}_0)$. Under Gaussian assumptions, it can be written as:

$$L_{\mathrm{DDPM}}(\epsilon_\theta) = \sum_{t=1}^{T} \mathbb{E}_{\boldsymbol{x}_0,\boldsymbol{\epsilon}_t}||\epsilon_\theta(\boldsymbol{x}_t, t) - \boldsymbol{\epsilon}_t||_2^2, \tag{1}$$

where $\epsilon_\theta$ is a parametrized noise estimator that predicts the noise added to $\boldsymbol{x}_t$. Song et al. (2020) proposes the denoising diffusion implicit model (DDIM) which is trained with the same objective (1), while the reverse process can be calculated deterministically.

## 3.2 EQUIVARIANCE AND EGNN

Satorras et al. (2021b) proposes the E(n)-equivariant EGNN graph neural network (EGNN), aiming to incorporate geometric symmetry into molecular modeling. Given a fully connected graph $\mathcal{G} = (v_i, e_{ij}), i \neq j$ where $e_{ij}$ are edges and $v_i$ are nodes with a coordinate $\boldsymbol{x}_i \in \mathbb{R}^3$ (*equivariant*) and a feature $\boldsymbol{h}_i \in \mathbb{R}^d$ (*invariant*), and a function $(\boldsymbol{x}', \boldsymbol{h}') = f(\boldsymbol{x}, \boldsymbol{h})$, the EGNN ensures the equivariance constraint (Appendix A.1). Namely, given an orthogonal matrix $\boldsymbol{R} \in \mathcal{R}^{3\times3}$ and a translation vector $\boldsymbol{t} \in \mathcal{R}^3$:

$$\boldsymbol{R}\boldsymbol{x}' + \boldsymbol{t}, \boldsymbol{h}' = f(\boldsymbol{R}\boldsymbol{x} + \boldsymbol{t}, \boldsymbol{h}).$$

i.e. $\boldsymbol{h}$ is invariant to transformations while $\boldsymbol{x}$ is receives the same transformation.

Hoogeboom et al. (2022) use the EGNN as noise predictors for the DDPM:

$$\hat{\boldsymbol{\epsilon}}_t^{(x)}, \hat{\boldsymbol{\epsilon}}_t^{(h)} = \epsilon_\theta(\boldsymbol{x}_t, \boldsymbol{h}_t, t) = \mathrm{EGNN}_\theta(\boldsymbol{x}_t, \boldsymbol{h}_t, t),$$

where $\hat{\boldsymbol{\epsilon}}_t^{(x)}, \hat{\boldsymbol{\epsilon}}_t^{(h)}$ are equivariant noise on $\boldsymbol{x}$ and invariant noise on $\boldsymbol{h}$, respectively, following the equivariance constraints above. This ensures the equivariance of the conditional probability $p(\boldsymbol{x}_{t-1}|\boldsymbol{x}_t)$. It is shown that with an invariant prior $p(\boldsymbol{x}_T)$, the *equivariant* process results in an *invariant* data distribution $p_\theta(\mathcal{X})$, where the probability of a data point remains the same after transformation (Köhler et al., 2020). This greatly improves data efficiency and model generalizability.

# 4 METHODS

## 4.1 SEMANTICS-GUIDED EQUIVARIANT DIFFUSION MODEL

### 4.1.1 SEMANTICS ENCODER

Let $(\boldsymbol{x}_0, \boldsymbol{h}_0)$ be an input 3D point cloud represented as a fully-connected graph and $(\boldsymbol{x}, \boldsymbol{h}) \in \mathcal{X}$ be any generic point from the data space $\mathcal{X}$. We design an encoder with an equivariant backbone that learns the conditional distribution of the semantics embedding $\boldsymbol{z}$ given the input (see Appendix A.4 for more details):

$$q(\boldsymbol{z}|\boldsymbol{x}_0) = \mathcal{N}(\boldsymbol{\mu}_{\boldsymbol{z}}, \boldsymbol{\sigma}_{\boldsymbol{z}}) \quad \boldsymbol{\mu}_{\boldsymbol{z}}, \boldsymbol{\sigma}_{\boldsymbol{z}} = \mathrm{Encoder}_\gamma(\boldsymbol{x}_0).$$

$\boldsymbol{z}$ is then sampled from the distribution and provided to a diffusion "decoder" to generate a reconstruction of the input. The embedding $\boldsymbol{z}$ is treated as a condition of the diffusion process.

Specifically, we only perform sampling of $\boldsymbol{z}$ from $q(\boldsymbol{z}|\boldsymbol{x}_0)$ during training. For the reconstruction, semantics-guided generation, and property manipulation experiments, we directly use $\boldsymbol{z} = \boldsymbol{\mu}_{\boldsymbol{z}}$.

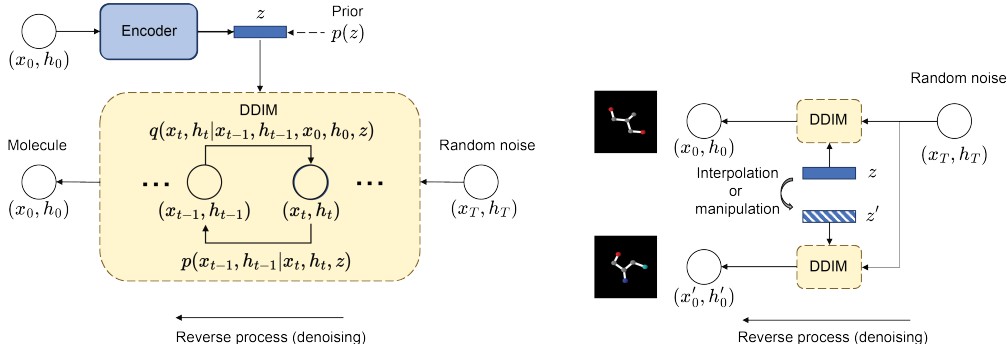

Figure 1: Left: the `egDiffAE` framework; the semantics embedding $z$ guides the generative denoising process. Right: the generative process for property manipulation and interpolation; the semantics embedding $z$ could be modified to control the content of the generation.

### 4.1.2 SEMANTICS-GUIDED DIFFUSION DECODER

The diffusion decoder $\epsilon_\theta$ predicts the amount of noise $\epsilon_t$ added to $(\boldsymbol{x}, \boldsymbol{h})$ at each time step in the forward process:

$$\hat{\boldsymbol{\epsilon}}_t^{(x)}, \hat{\boldsymbol{\epsilon}}_t^{(h)} = \epsilon_\theta(\boldsymbol{x}_t, \boldsymbol{h}_t, \boldsymbol{z}, t) = \text{EGNN}_\theta(\boldsymbol{x}_t, \boldsymbol{h}_t, \boldsymbol{z}, t). \tag{2}$$

Thus, the diffusion objective becomes:

$$L_D(\epsilon_\theta) = \sum_{t=1}^{T} \mathbb{E}_{(\boldsymbol{x}_0, \boldsymbol{h}_0), \boldsymbol{z}, \boldsymbol{\epsilon}_t^{(x)}, \boldsymbol{\epsilon}_t^{(h)}} [||\hat{\boldsymbol{\epsilon}}_t^{(x)} - \boldsymbol{\epsilon}_t^{(x)}||_2^2 + ||\hat{\boldsymbol{\epsilon}}_t^{(h)} - \boldsymbol{\epsilon}_t^{(h)}||_2^2], \tag{3}$$

where $\boldsymbol{\epsilon}_t^{(x)}, \boldsymbol{\epsilon}_t^{(h)} \sim \mathcal{N}(0, I)$. This is basically the same as (1) but the noise estimator $\epsilon_\theta$ takes both $(\boldsymbol{x}_t, \boldsymbol{h}_t)$ and the semantics embedding $\boldsymbol{z}$ as inputs. Intuitively, here $\boldsymbol{z}$ can be considered as controlling the "direction" of the denoising (i.e. generation) towards the desired semantics. In 4.2, we also enforce the maximal mutual information (MI) between $\boldsymbol{z}$ and the input, which empowers $\boldsymbol{z}$ to effectively guide and control the generation processes.

Since deterministic reconstruction is needed for an autoencoder-like framework, we use the DDIM sampling (Song et al., 2020). Starting from the random noise $\boldsymbol{x}_T$, we could obtain a reconstruction of $\boldsymbol{x}_0$ by progressively removing the predicted noise in a deterministic manner:

$$\begin{aligned}
\boldsymbol{x}_{t-1} &= \sqrt{\alpha_{t-1}} \left( \frac{\boldsymbol{x}_t - \sqrt{1 - \alpha_t} \hat{\boldsymbol{\epsilon}}_t^{(x)}}{\sqrt{\alpha_t}} \right) + \sqrt{1 - \alpha_{t-1}} \hat{\boldsymbol{\epsilon}}_t^{(x)}, \\
\boldsymbol{h}_{t-1} &= \sqrt{\alpha_{t-1}} \left( \frac{\boldsymbol{h}_t - \sqrt{1 - \alpha_t} \hat{\boldsymbol{\epsilon}}_t^{(h)}}{\sqrt{\alpha_t}} \right) + \sqrt{1 - \alpha_{t-1}} \hat{\boldsymbol{\epsilon}}_t^{(h)}.
\end{aligned} \tag{4}$$

As shown in Song et al. (2020) and Dhariwal & Nichol (2021), this process can in turn be reversed when the time step is sufficiently small. Thus, we could both map a noise point $(\boldsymbol{x}_T, \boldsymbol{h}_T)$ to a data point $(\boldsymbol{x}_0, \boldsymbol{h}_0)$ and vice versa. We use the deterministically mapped noise point for the interpolation and property manipulation experiments.

### 4.2 REGULARIZATION OF THE SEMANTIC EMBEDDING

To control the scale and shape of $\boldsymbol{z}$, we employ a regularization term on the marginal distribution $q(\boldsymbol{z}) = \int_{\boldsymbol{x}_0} q_\gamma(\boldsymbol{z}|\boldsymbol{x}_0) q(\boldsymbol{x}_0) \mathrm{d}\boldsymbol{x}_0$. Specifically, we use a sample-based kernel maximum mean discrepancy (MMD) (Gretton et al., 2012) on mini-batches of size $n$ to make $q(\boldsymbol{z})$ approach the shape of a Gaussian prior $p(\boldsymbol{z}) = \mathcal{N}(0, I)$:

$$\text{MMD}(q(\boldsymbol{z})||p(\boldsymbol{z})) = \frac{1}{n^2} \left[ \sum_{i \neq j} k(\boldsymbol{z}_i, \boldsymbol{z}_j) + \sum_{i \neq j} k(\boldsymbol{z}_i', \boldsymbol{z}_j') - 2 \sum k(\boldsymbol{z}_i, \boldsymbol{z}_j') \right], 1 \leq i, j \leq n,$$

where $k$ is the kernel function, $z_i$'s are obtained from the data points in the minibatch and $z_i'$'s are randomly sampled from $p(z) = \mathcal{N}(0, I)$. This objective can be effectively calculated from the sample.

Our full training objective is as follows:

$$L = L_D(\epsilon_\theta) + \beta \text{MMD}(q(z)||p(z)), \tag{5}$$

where $\beta > 0$ is the regularization strength. The full model schematic is shown in Figure 1.

### 4.3 THEORETICAL PERSPECTIVE OF THE OBJECTIVE

One concern with the diffusion autoencoder is that the decoder may ignore the embedding $z$ and solely rely on the noise $x_T$ for the generation; this mirrors a similar problem observed in several generative models (Bowman et al., 2015; Che et al., 2016). We show that this objective also ensures maximal MI between $z$ and the input $x_0$. It has been shown that maximizing MI ensures informative embeddings in geometric molecular modeling (Liu et al., 2022; 2023).

For clarity, here we use one character $x$ to denote a point in the data space $\mathcal{X}$. When $\beta = 1$ (proof in Appendix A.2):

$$L = -L_{\text{ELBO}} - \text{MI}(x_0, z) \tag{6}$$
$$L_{\text{ELBO}} := -L_D(\epsilon_\theta) - \mathbb{E}_{q(x_0)} \text{KL}(q(z|x_0)||p(z)).$$

KL is the Kullback-Leibler divergence. $L_{\text{ELBO}}$ is equivalent to the negative ELBO for $\log p_\theta(x)$ under our model assumption. $L_D(\epsilon_\theta)$ is the same objective in (3). This means that by minimizing $L$, we could jointly maximize the ELBO and the MI.

In practice, since $\text{KL}(q(z)||p(z))$ is intractable, we need to approximate it with an MMD. Both the KL divergence and the MMD are minimized when $q(z) = p(z)$, thus justifying the approximation. This leads to our objective function (5).

## 5 EXPERIMENTS

### 5.1 SETUP

We empirically evaluate our method using three benchmarking datasets (see Appendix A.3 for more details on data processing):

**QM9** (Ramakrishnan et al., 2014) is one gold-standard dataset for machine learning on 3D molecules. It contains molecules comprised of [H, C, N, O, F] and $\leq 9$ heavy atoms. Each molecule has a 3D conformation paired with several quantum chemical properties.

**ZINC** (Sterling & Irwin, 2015) includes larger, drug-like molecules with the quantitative estimate of drug-likeness (QED) scores (Bickerton et al., 2012).

**GEOM** (Axelrod & Gomez-Bombarelli, 2022) includes annotated conformers for 450,000 molecules with an average of 44 atoms (25 heavy atoms).

The 3D molecular structure is represented as a fully connected graph with atoms as nodes. Each node carries its atomic type $h$ and 3D coordinate $x$. Following Xu et al. (2022) and Hoogeboom et al. (2022), we remove the center of mass for the coordinate to ensure translational invariance. We only include the heavy atoms. According to Hoogeboom et al. (2022), taking hydrogen into account does not lead to better generation, but poses more challenges as the scale of the data is increased $\sim 3$ fold.

### 5.2 EVALUATION METRICS

For evaluation of the generated molecules, we use the stability metric as in Satorras et al. (2021a); Hoogeboom et al. (2022); Xu et al. (2022). The bonds and their orders are inferred by comparing the atomic distances with the standard range of bond distances. We report the proportion of valid, unique and novel molecules in the generated set with the same metrics as in Vignac et al. (2023), as well as the estimated molecular energy for the generated molecules (more details in Appendix A.7).

Furthermore, the bond lengths are much more irregular in larger and more complicated molecules, making bond order inference difficult, as observed in Hoogeboom et al. (2022); Xu et al. (2023). Thus, we follow the practice in these studies and only report atomic stability for ZINC and GEOM.

For the comparison of molecular similarity, we use the cosine similarity between the semantic embeddings from a separate egDiffAE model (Appendix A.7.1)

## 5.3 3D SEMANTICS-GUIDED GENERATION

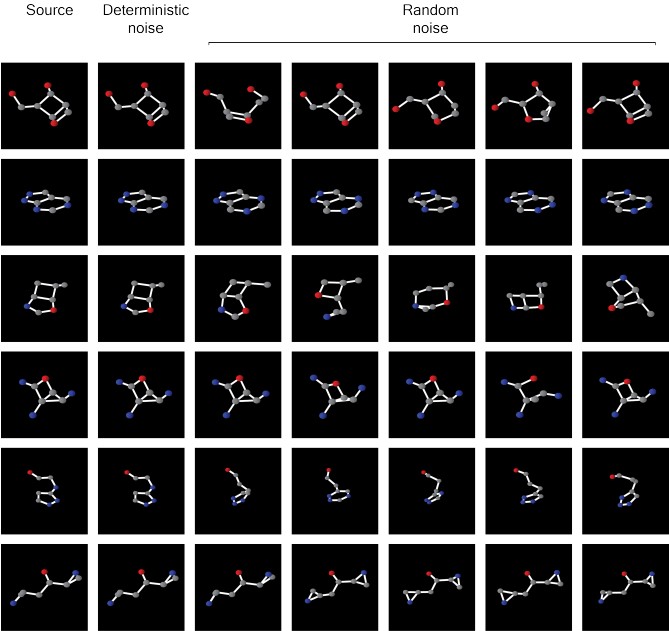

Figure 2: QM9 semantics-guided generation. Molecules in the same row are generated with the same $z$ but different noise. Note that the molecules are superimposed on the source for better comparison, but the actual generations have arbitrary orientations due to random $x_T$.

semantics-guided generation uses the embedding $z$ from the "source molecule" combined with determinstically mapped or randomly sampled noise $(x_T, h_T)$. We compare our methods with the following baselines:

- mpDiffAE: the same framework as egDiffAE but using a non-equivariant message-passing diffusion decoder. The coordinates are simply concatenated to other node features.
- egDDIM: a standard equivariant DDIM with a similar architecture as in Hoogeboom et al. (2022) where there is no semantic guidance, so the results are not directly comparable with the rest.
- raw: the evaluation metrics on the raw test dataset.

Table 1: Semantics-guided generation results on QM9. Note that egDDIM is not directly comparable here as its generation does not involve any guidance.

|  | %valid | %unique ↑ | %novel ↑ | similarity ↑ | energy ↓ |
|---|---|---|---|---|---|
| egDiffAE | 0.82 | 0.57 | 0.58 | 0.97±0.03 | -0.43±4.23 |
| mpDiffAE | 0.68 | 0.45 | 0.54 | 0.90±0.11 | 0.24±3.89 |
|  |  |  |  |  |  |
| egDDIM (no guidance) | 0.88 | 0.85 | 0.43 | - | -0.82±3.03 |
| raw | 0.97 | 1.00 | 1.00 | - | -2.17±2.10 |

As expected, we could achieve near-perfect reconstruction with deterministically mapped noise (Figure 2 second column; Appendix Table 5).

With random noise instead of deterministic ones, we are able to explore the neighborhood of the semantics source molecule in the data distribution. For each of the 1000 samples selected from the test set, we generate 10 random molecules using random noise and the exact $z$ of the source. As shown in Figure 2 (third to last columns), using random $(x_T, h_T)$ and the $z$ of the template molecule, the generated molecules have overall similar structures and compositions as the semantics source with only minor random variations. This shows that the semantic embedding dominates the generation process and carries most of the important information about the source molecule, as intended. Furthermore, the generation quality of egDiffAE is much higher than the non-equivariant mpDiffAE due to its greater data efficiency (Table 1; results for ZINC and GEOM in Appendix A.8 and Table 6).

## 5.4  INTERPOLATION BETWEEN 3D MOLECULE PAIRS

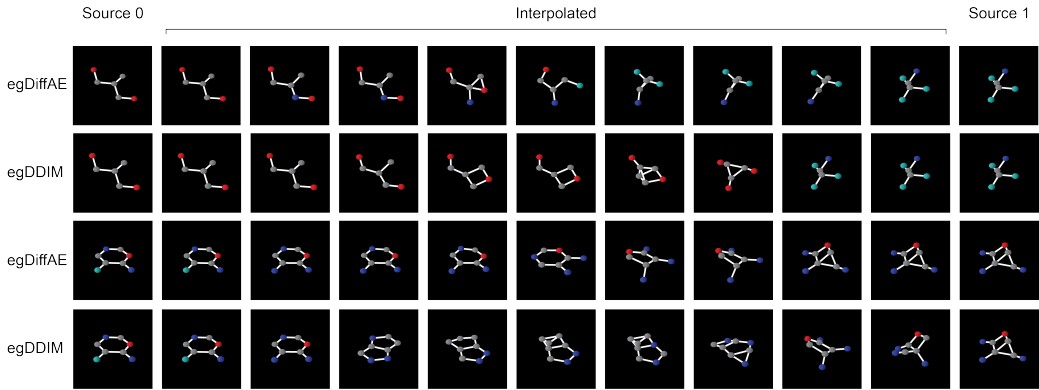

Figure 3: Interpolation examples on two randomly selected molecule pairs. All molecules are superimposed on their respective semantics source.

Table 2: Interpolation results. We report cosine similarity-based measurements of the smoothness of the interpolation trajectory.

| Model | Smoothness mean ↑ | Smoothness std ↓ | Midpoint similarity ↑ |
|---|---|---|---|
| egDiffAE | 0.87±0.07 | 0.15±0.08 | 0.57±0.19 |
| egDDIM | 0.75±0.10 | 0.24±0.09 | 0.24±0.27 |
| random | 0.06±0.14 | 0.33±0.08 | 0.04±0.26 |

For any pair of real molecules $(x_0^{(i)}, h_0^{(i)})$ and $(x_0^{(j)}, h_0^{(j)})$, we obtain their corresponding noise $(x_T^{(i)}, h_T^{(i)})$ and $(x_T^{(j)}, h_T^{(j)})$ from back-sampling, and semantic embedding $z^{(i)}$ and $z^{(j)}$ from the encoder. For $\lambda \in [0, 1]$ we perform linear interpolation for both the noise and the embedding:

$$x_T^{(\lambda)} = \lambda x_T^{(i)} + (1 - \lambda)x_T^{(j)}, h_T^{(\lambda)} = \lambda h_T^{(i)} + (1 - \lambda)h_T^{(j)}$$
$$z^{(\lambda)} = \lambda z^{(i)} + (1 - \lambda)z^{(j)}.$$

We then generate new molecules $(x_0^{(\lambda)}, h_0^{(\lambda)})$ from $(x_T^{(\lambda)}, h_T^{(\lambda)})$ and $z^{(\lambda)}$. For egDDIM, only the noise is interpolated. We evaluate the interpolation results on 1000 random pairs (10 interpolated samples for each pair with $\lambda \in [0, 1]$) for egDiffAE and egDDIM. For each interpolation trajectory, we calculate three metrics (similarity metrics are the same as in 5.2):

- Smoothness mean and std: "smoothness" is measured by the similarity between adjacent steps. We then calculate the mean and standard deviation of the smoothness score for all adjacent steps in the trajectory. The former reflects the average smoothness and the latter measures whether the trajectory takes uniform steps or sharp turns.

- Midpoint similarity: the similarity between the midpoint of the trajectory and its two endpoints. This score measures how much the interpolation process preserves the information from the endpoints. Intuitively, the interpolated sample should be a mixture of both, rather than a vastly different one.

All scores are then averaged across the 1000 trajectories. We also include a random baseline where the metrics are calculated on trajectories of the same size but with random samples from the test set. As shown in Table 2 and Figure 3, our model is able to achieve smoother interpolations than the standard egDDIM which has no semantic guidance.

## 5.5  3D SEMANTICS-GUIDED MOLECULAR PROPERTY MANIPULATION

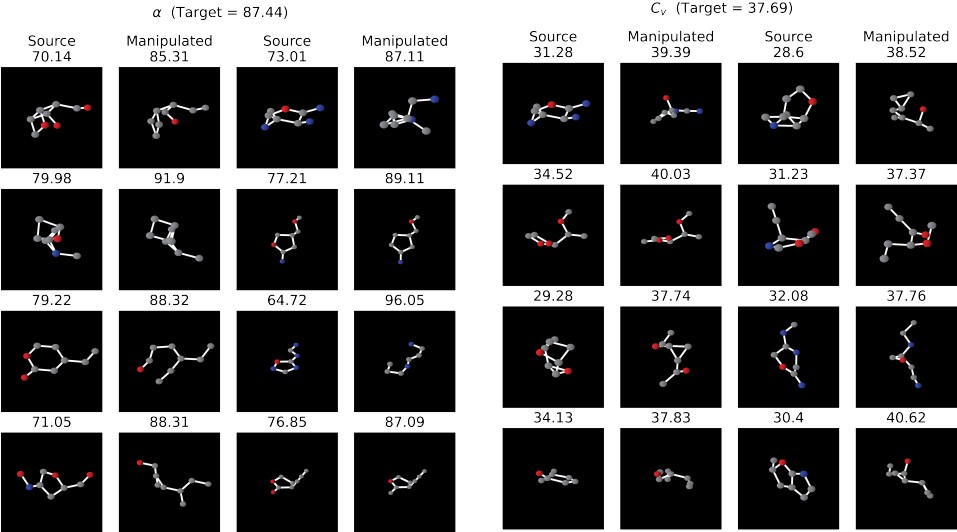

Figure 4: Property manipulation examples. The property values of the source (left columns) and the manipulated molecule (right columns) are shown.

Table 3: Property manipulation results. We report the number of "success" cases filtered by MAE, validity and similarity with the source. raw and egDiffAE-retrieval provide a lower bound and an approximate upper bound, respectively. More details in Appendix Table 8.

| Mode | Method | %success ↑ (valid, MAE$< 0.5\sigma_p$, similarity$> 0.5$) | | | | | |
| | | $\mu$ | $\alpha$ | $\epsilon_{HOMO}$ | $\epsilon_{LUMO}$ | $\Delta\epsilon$ | $C_v$ |
|---|---|---|---|---|---|---|---|
| Manipulated molecules | egDiffAE | 0.09 | 0.38 | 0.20 | 0.36 | 0.30 | 0.27 |
| | mpDiffAE | 0.03 | 0.07 | 0.00 | 0.01 | 0.01 | 0.09 |
| | backprop | 0.00 | 0.00 | 0.00 | 0.00 | 0.00 | 0.00 |
| Known molecules | (L) raw | 0.11 | 0.12 | 0.1 | 0.21 | 0.18 | 0.15 |
| | (U) egDiffAE-retrieval | 0.15 | 0.31 | 0.17 | 0.43 | 0.37 | 0.31 |

Here we aim to obtain molecules that are sufficiently similar to the semantics source yet with altered properties. We use a simple linear property manipulation framework on the semantic embedding: we train a linear regression on the semantic embeddings to predict the property. Given an embedding $z$, target value $y'$, and the weight and bias of the linear regression $(w, b)$, we can obtain a new embedding with the desired property via: $s = (y' - b - z^T w)/w^T w$; $z' = z + sw$ (see Appendix A.9 for more details). We then use the manipulated embedding $z'$ to obtain a new molecule via two different methods:

- egDiffAE: $z'$ is fed to the diffusion decoder to generate a manipulated molecule with deterministic noise.
- egDiffAE-retrieval: we "retrieve" a molecule in the training set whose embedding has the highest cosine similarity with $z'$. Note this does not generate any novel molecules, so it's not practically useful, but serves as an approximate performance upper bound.

As comparable methods for this task are scarce, we design the following baselines:

- mpDiffAE: the same model from 5.3, with the same manipulation method as egDiffAE.

- `backprop`: we backpropagate the classifier loss towards the target value back to the input.
- `raw`: the evaluation metrics on the raw data, which is the lower bound of the performance.

The experiment is performed on each property individually. The property values of the generated molecules are predicted with an externally trained EGNN-based classifier (Appendix A.4). We evaluate the manipulated molecules by their validity, mean absolute error (MAE) to the target value, and similarity to the semantics source. A molecule is considered "success" if it is valid, has MAE $< 0.5\sigma_p$ ($\sigma_p$ is the standard deviation of the property), and has similarity $> 0.5$. Table 3 shows that `egDiffAE` outperforms the baselines, almost competitive with the approximate upper bound (`egDiffAE-retrieval`).

We also observe that the generated molecules generally respect the original structure of the source with high similarity (Figure 4; Appendix Table 8). Overall, the embedding can be efficiently used to ensure both semantic similarity with the source and the target property value.

## 5.6 ABLATION STUDY

Table 4: Ablation study.

| | Semantics-guided generation | | | | | Property manipulation %success ↑ | | |
| | %valid ↑ | %unique ↑ | %novel ↑ | similarity ↑ | energy ↓ | $\mu$ | $\alpha$ | $C_v$ |
|---|---|---|---|---|---|---|---|---|
| base | 0.82 | 0.57 | 0.58 | 0.97±0.03 | -0.43±4.23 | 0.09 | 0.38 | 0.27 |
| prob- | 0.77 | 0.54 | 0.56 | 0.97±0.02 | -0.41±5.13 | 0.1 | 0.38 | 0.34 |
| MMD- | 0.75 | 0.48 | 0.58 | 0.97±0.02 | -0.60±4.22 | 0.08 | 0.4 | 0.33 |

The ablation study is performed on QM9. We compare our base model with one that removes the MMD loss (`MMD-`) and one that uses a non-probabilistic encoder (`prob-`), where $z$ is deterministically calculated instead of randomly sampled during training. We find that in either case, the semantics-guided generation quality becomes much lower than the base model, indicating a less regularized latent space (Table 4, left).

The performance of property manipulation does not differ much across the models, except that the full model has generally higher validity rates. This is mainly because the task only searches the close proximity of known $z$'s. Thus, the regularization of $z$'s distribution should not have a strong impact on the results (Table 4, right; full results in Appendix A.10).

## 6 CONCLUSION AND OUTLOOK

In this paper we present `egDiffAE`, which can effectively learn meaningful latent semantic embeddings from 3D molecules, and subsequently leverage these embeddings to guide the generative denoising process in diffusion models. The resulting embedding controls most of the contents of the generation, which enables easy operations on such embeddings to effectively explore the 3D molecular space. Our regularization of the embedding space further improves the generation quality. We show, for example, these embeddings facilitate semantics-guided generative tasks in diffusion models such as interpolation and property manipulation; we empirically verify our `egDiffAE` approach, demonstrating its superior performance compared to the baselines.

We believe that by incorporating some recent progress in large 3D molecule generation, our framework could be further extended to larger molecules and more complex properties such as intermolecular interactions, with potential applications in drug design and discovery. We also acknowledge that our semantics-guided generative experiments are mostly limited to alterations of a small number of atoms and their relative positions. In the application scenarios that require drastic changes, such as the addition or deletion of whole functional groups, more informative semantic embeddings and careful regularizations are needed. We could also potentially leverage both semantics guidance and external conditioning (such as the structure pf receptor pockets or molecular fragments) to achieve conditioned design with finer control on the compositions, structures, properties, etc. of the generation molecules. We leave these for future investigations.

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

# A  APPENDIX

## A.1  THE EGNN ARCHITECTURE

Following Satorras et al. (2021b), given a fully connected graph $\mathcal{G} = (v_i, e_{ij}), i \neq j$ where $e_{ij}$ are edges and $v_i$ are nodes with a coordinate $\boldsymbol{x}_i \in \mathbb{R}^3$ (*equivariant*) and a feature $\boldsymbol{h}_i \in \mathbb{R}^d$ (*invariant*), the EGNN convolution layer (EGCL) is as follows:

$$\boldsymbol{m}_{ij} = \phi_e(\boldsymbol{h}_i^l, \boldsymbol{h}_j^l, d_{ij}^2, a_ij)$$

$$\boldsymbol{h}_i^{l+1} = \phi_h(\boldsymbol{h}_i^l, \sum_{j \neq i} e_{ij}\boldsymbol{m}_{ij})$$

$$\boldsymbol{x}_i^{l+1} = \boldsymbol{x}_i^l + \sum_{j \neq i} \frac{\boldsymbol{x}_i^l - \boldsymbol{x}_j^l}{d_{ij} + 1}\phi_x(\boldsymbol{h}_i^l, \boldsymbol{h}_j^l, d_{ij}^2, a_{ij}),$$

where $\phi_e, \phi_h, \phi_x$ are functions with learnable parameters and $d_{ij} = ||x_i^l - x_j^l||_2$.

## A.2  PROOF OF 4.3

For clarity, in this section we use one character $\boldsymbol{x}$ to denote a point in the data space $\mathcal{X}$. When conditioned on $\boldsymbol{z}$, the negative ELBO becomes:

$$-\mathbb{E}_{q(\boldsymbol{x}_0)} \log p(\boldsymbol{x}_0) \leq -\mathbb{E}_{q(\boldsymbol{x}_0)}\mathbb{E}_{q(\boldsymbol{z}|\boldsymbol{x}_0)} \log \frac{p(\boldsymbol{x}_0, \boldsymbol{z})}{q(\boldsymbol{z}|\boldsymbol{x}_0)}$$

$$\leq -\mathbb{E}_{q(\boldsymbol{x}_0)}\mathbb{E}_{q(\boldsymbol{z}|\boldsymbol{x}_0)}\mathbb{E}_{q(\boldsymbol{x}_{1:T}|\boldsymbol{x}_0)} \log \frac{p(\boldsymbol{x}_{0:T}, \boldsymbol{z})}{q(\boldsymbol{x}_{1:T}|\boldsymbol{x}_0)q(\boldsymbol{z}|\boldsymbol{x}_0)}$$

$$= -\mathbb{E}_{q(\boldsymbol{x}_0)}\left[L_0 + \sum_{t=2}^{T} L_t + L_T + \mathrm{KL}(q(\boldsymbol{z}|\boldsymbol{x}_0)||p(\boldsymbol{z}))\right],$$

where

$$L_0 = -\mathbb{E}_{q(\boldsymbol{x}_1|\boldsymbol{x}_0)}\mathbb{E}_{q(\boldsymbol{x}_1|\boldsymbol{x}_0)} \log p_\theta(\boldsymbol{x}_0|\boldsymbol{x}_1, \boldsymbol{z}), L_T = \mathrm{KL}(q(\boldsymbol{x}_T|\boldsymbol{x}_0)||p(\boldsymbol{x}_T))$$

$$L_t = \mathbb{E}_{q(\boldsymbol{x}_t|\boldsymbol{x}_0)}\mathbb{E}_{q(\boldsymbol{z}|\boldsymbol{x}_0)}\mathrm{KL}(q(\boldsymbol{x}_{t-1}|\boldsymbol{x}_t, \boldsymbol{x}_0)||p(\boldsymbol{x}_0|\boldsymbol{x}_1, \boldsymbol{z})).$$

As in Ho et al. (2020), we ignore the constant $L_T$ and the border case $L_0$, and use the reparametrization $\boldsymbol{x}_t = \sqrt{\alpha_t}\boldsymbol{x}_0 + \sqrt{1 - \alpha_t}\boldsymbol{\epsilon}_t, \boldsymbol{\epsilon}_t \sim \mathcal{N}(0, I)$. Then the RHS can be simplified into:

$$\sum_{t=1}^{T} \mathbb{E}_{\boldsymbol{x}_0, \boldsymbol{z}, \boldsymbol{\epsilon}_t}||\epsilon_\theta(\boldsymbol{x}_t, \boldsymbol{z}, t) - \boldsymbol{\epsilon}_t||_2^2 + \mathbb{E}_{q(\boldsymbol{x}_0)}\mathrm{KL}(q(\boldsymbol{z}|\boldsymbol{x}_0)||p(\boldsymbol{z})) \tag{7}$$

$$= L_D(\epsilon_\theta) + \mathbb{E}_{q(\boldsymbol{x}_0)}\mathrm{KL}(q(\boldsymbol{z}|\boldsymbol{x}_0)||p(\boldsymbol{z})) := -L_{\mathrm{ELBO}}.$$

$L_D(\epsilon_\theta)$ is the same as (3). This is almost the same form as the DDPM objective (1), except for the condition $\boldsymbol{z}$ and the KL divergence term.

By adding a KL divergence regularizer between $q(\boldsymbol{z})$ and $p(\boldsymbol{z})$:

$$L = L_D(\epsilon_\theta) + \beta\mathrm{KL}(q(\boldsymbol{z})||p(\boldsymbol{z}))$$

$$= L_D(\epsilon_\theta) + \beta[\mathbb{E}_{q(\boldsymbol{x}_0)}\mathrm{KL}(q(\boldsymbol{z}|\boldsymbol{x}_0)||p(\boldsymbol{z})) - \mathrm{MI}(\boldsymbol{x}_0, \boldsymbol{z})]$$

when $\beta = 1$:

$$L = L_D(\epsilon_\theta) + \mathbb{E}_{q(\boldsymbol{x}_0)}\mathrm{KL}(q(\boldsymbol{z}|\boldsymbol{x}_0)||p(\boldsymbol{z})) - \mathrm{MI}(\boldsymbol{x}_0, \boldsymbol{z}) \tag{8}$$

$$= -L_{\mathrm{ELBO}} - \mathrm{MI}(\boldsymbol{x}_0, \boldsymbol{z})$$

## A.3  DATA PROCESSING

**QM9**  We exclude molecules with $< 2$ heavy atoms. The final dataset has 133,877 data points.

**ZINC**   As the dataset has no 3D information, we use the lowest energy conformation predicted by Rdkit (Landrum et al., 2016). We only keep molecules with $\leq 20$ heavy atoms, resulting in 69,917 data points.

**GEOM**   We only use molecules with $\leq 40$ heavy atoms and discard molecules containing heavy atoms other than [C, N, O, F, Cl, Br, P, S] to reduce the number of features, because all other atom types are very rare. For each molecule, we include its 30 lowest energy conformers. The final dataset has 6,876,297 data points.

The datasets are split into train/test/val by 8:1:1.

## A.4   MODEL ARCHITECTURE

The diffusion decoder takes the noisy molecular graph at time $t$ and the semantics embedding $z$ to estimate the noise:

$$\hat{\boldsymbol{\epsilon}}_t^{(x)}, \hat{\boldsymbol{\epsilon}}_t^{(h)} = \epsilon_\theta(\boldsymbol{x}_t, \boldsymbol{h}_t, \boldsymbol{z}, t) = \text{EGNN}_\theta(\boldsymbol{x}_t, \boldsymbol{h}_t, \boldsymbol{z}, t),$$

where $t$ is represented by a sinusoidal embedding. Both $t$ and $\boldsymbol{z}$ are concatenated to the node features as the input to the EGNN.

The encoder is another EGNN that takes the molecular graph as input and transforms it into a $d$-dimensional invariant embedding and the equivariant embedding:

$$\boldsymbol{z}^{(x)}, \boldsymbol{z}^{(h)} = \text{EGNN}(\boldsymbol{x}_0, \boldsymbol{h}_0).$$

For each EGNN layer, we take the sum of all node embeddings. We then concatenate the average embedding from all layers and transform them into a single vector through a linear layer to obtain the semantic embedding.

$$\tilde{\boldsymbol{z}}^{(h),l} = \frac{1}{n_{\max}} \sum_{j=1}^{n} \boldsymbol{z}^{(h),l,j}$$

$$\boldsymbol{\mu}_z, \boldsymbol{\sigma}_z = \text{linear}(\text{cat}(\tilde{\boldsymbol{z}}^{(h),0}, \tilde{\boldsymbol{z}}^{(h),1}, ..., \tilde{\boldsymbol{z}}^{(h),L})),$$

where $l = 1, 2, ..., L$ is the number of layers, $n$ is the number of nodes, and $n_{\max}$ is the maximum number of nodes, which is used as the regularizer of the output scale.

For the evaluation of property manipulation results, we also train an EGNN-based classifier that uses the output embedding to predict the six properties listed in Table 3.

## A.5   MODEL TRAINING AND SAMPLING DETAILS

In our experiments, the diffusion decoder has 9 EGNN layers for QM9 and ZINC, and 4 for GEOM. The encoders have 2 EGNN layers. All EGNN layers have 256 hidden dimensions. The dimension of semantic embeddings is 16. We fix the log variance of the encoder to $-3$. The $n_{\max}$ is 20 for QM9 and ZINC, and 40 for GEOM. The time point representation $t$ uses 16-dimensional sinusoidal embedding.

For the models in comparison, `mpDiffAE` has the same number of layers and dimensions, but without the layers are non-equivariant message-passing layers. `egDDIM` is a standard equivariant diffusion model which does not use any semantics embedding.

All diffusion models have 1000 diffusion steps and a linear noise schedule where $\beta_1 = 0.0001$ and $\beta_T = 0.02$. All models are trained on 1 RTX3090 GPU with a batch size of 128, 500 epochs for QM9 and ZINC and 10 epochs for GEOM, taking $\sim$2 days.

The property classifier has 4 EGNN layers (32 hidden dimensions) and a 2-layer MLP (64 hidden dimensions) to predict the properties (6 dimensions), trained with a batch size of 128 and 500 epochs.

For the DDIM sampling, we always use 100 denoising steps (i.e. with a stride of 10 time points), which means the forward process time trajectory is $[0, 1, ..., 1000]$ while the denoising (reverse)

process time trajectory is $[1000, 900, ..., 0]$. The same stride is used to reverse map $(\boldsymbol{x}_0, \boldsymbol{h}_0)$ deterministically to the noise point $(\boldsymbol{x}_T, \boldsymbol{h}_T)$.

## A.6 DETERMINISTIC RECONSTRUCTION

Table 5: Reconstruction

|  | QM9 | | QM9 (10) | | ZINC | | GEOM | |
| --- | --- | --- | --- | --- | --- | --- | --- | --- |
|  | $\boldsymbol{h}$ acc ↑ | $\boldsymbol{x}$ MSE ↓ | $\boldsymbol{h}$ acc ↑ | $\boldsymbol{x}$ MSE ↓ | $\boldsymbol{h}$ acc ↑ | $\boldsymbol{x}$ MSE ↓ | $\boldsymbol{h}$ acc ↑ | $\boldsymbol{x}$ MSE ↓ |
| egDiffAE | 1.000 | 0.014 | 0.998 | 0.201 | 1.000 | 0.018 | 1.000 | 0.036 |
| mpDiffAE | 1.000 | 0.020 | 1.000 | 0.137 | 1.000 | 0.027 | 1.000 | 0.028 |
| egDDIM | 1.000 | 0.022 | 0.967 | 0.465 | 0.999 | 0.053 | 1.000 | 0.135 |

The DDIM framework allows deterministic mapping of the data point $(\boldsymbol{x}_0, \boldsymbol{h}_0)$ to a noise point $(\boldsymbol{x}_T, \boldsymbol{h}_T)$, and then deterministic reconstruction, unlike in DDPM where the generation is stochastic. For `egDiffAE` and `mpDiffAE` we use the $\boldsymbol{z}$ and the deterministic $(\boldsymbol{x}_T, \boldsymbol{h}_T)$ of the input for the reconstruction, and for `egDDIM` only $(\boldsymbol{x}_T, \boldsymbol{h}_T)$. The reconstructions are evaluated by the accuracy of node features (atom types) and the MSE of coordinates.

As expected, all diffusion model-based methods can achieve near-perfect reconstruction (Table 5; Figure 2 second column). In this application, `mpDiffAE` is comparable with `egDiffAE`, because the entire process is deterministic as long as the noises are sufficiently small, and the orientation of the molecule are pre-determined. Interestingly, we notice that if we use only 10 denoising steps (i.e. with a stride of 100 time points; Table 5 "QM9 (10)" column), the performance of `egDiffAE` and `mpDiffAE` is slightly worse while that of `egDDIM` drops significantly. This shows that the semantic vector $\boldsymbol{z}$ does encode sufficient information and can be used to accelerate the denoising process.

## A.7 DETAILS FOR EVALUATION METRICS

### A.7.1 SIMILARITY

To our knowledge, it's difficult to compare the similarity of 3D molecules. In this study, we use the semantic embedding from egDiffAE with a non-probabilistic encoder for the comparison. The model is trained on the same training set with 8-dimensional semantic embeddings for QM9 and 16 for ZINC and GEOM. The similarity is then defined as the cosine similarity between the embeddings.

### A.7.2 ENERGY ESTIMATION

The energy of the molecules is calculated using the xTB package (python interface) (Bannwarth et al., 2019). As we are modeling with implicit hydrogen, hydrogen atoms with their coordinates are first added to the molecule using RDKit. The atom types and coordinates are then passed to the single-point calculator (GFN2xTB) with default parameters to obtain the energy. We show in Figure 5A that for QM9, the energy estimation is almost the same with real hydrogen coordinates and with hydrogen added by RDKit. Thus, we use RDKit to estimate hydrogen coordinates for all generated molecules. We report the average energy among valid molecules. As this calculation would require valid molecules, it does not apply to ZINC and GEOM since the generations are mostly disconnected.

Figure 5B shows the energy distribution of molecules from semantics-guided generation and Figure 5C property manipulation. In general, we see a slight shift towards higher energy than the data distribution. For $\epsilon_{HOMO}$, $\epsilon_{LUMO}$ and $\Delta\epsilon$, where the classification from the semantic is more difficult (see A.9), we see a greater shift.

### A.7.3 UNIQUENESS AND NOVELTY

Following Vignac et al. (2023), a generated molecule is considered "novel" if its canonical SMILES is not observed in the training set, and "unique" if its SMILES only appears once in the generation

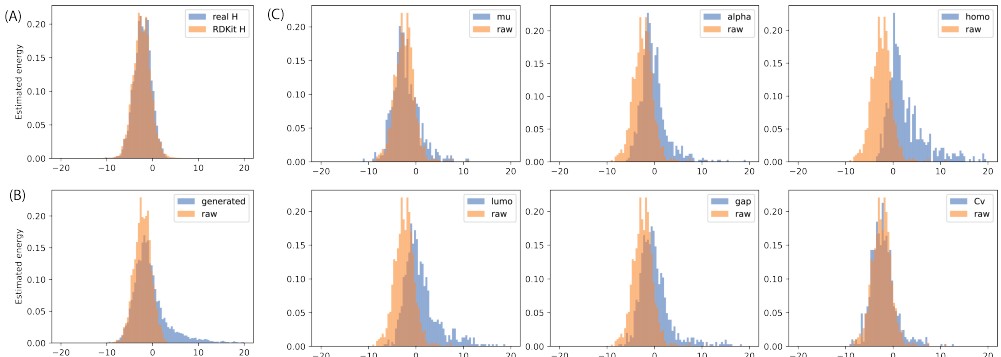

Figure 5: Distribution of energy for (A) real molecules with hydrogen estimated by xTB with real hydrogen coordinates and those estimated by RDKit; (B) semantics-guided generated molecules; (C) property-manipulated molecules.

set. We'd like to point out that these metrics are not particularly meaningful for the semantics-guided generation because of the setting: (1) it is desired that molecules generated by the same semantics embedding should be similar, or even identical, naturally resulting in lower uniqueness; (2) as we are using the test set as the semantics source, the generated molecules are very likely to differ from the training set, in turn naturally resulting in higher novelty.

## A.8 EXTENDED RESULTS FOR GENERATION

Figure 6 shows sample semantics-guided generations for ZINC and QM9. We observe that with no explicit modeling of the bonds, generation on large molecules usually ends up with disconnected fragments, similar to the observation in Hoogeboom et al. (2022); Xu et al. (2023). This makes validity and energy measurement meaningless. Thus, we only report the atom stability and similarity to the source in Table 6. The generation quality could potentially be improved by including the 2D molecular graph into the modeling (Vignac et al., 2023).

Table 6: Semantics-guided generation results

|  | ZINC | | GEOM | |
|---|---|---|---|---|
|  | %stable ↑ | similarity ↑ | %stable ↑ | similarity ↑ |
| egDiffAE | 0.93 | 0.93±0.03 | 0.71 | 0.95±0.03 |
| mpDiffAE | 0.94 | 0.56±0.15 | 0.67 | 0.60±0.15 |
| egDDIM (no guidance) | 0.88 | - | 0.69 | - |
| raw | 0.99 | - | 0.79 | - |

Table 7: *De novo* generation results

|  | QM9 | | | | ZINC | GEOM |
|---|---|---|---|---|---|---|
| Model | %valid ↑ | %novel ↑ | %unique ↑ | energy ↓ | %stable ↑ | %stable ↑ |
| egDiffAE | 0.66 | 0.45 | 0.62 | 1.21±5.42 | 0.90 | 0.65 |
| mpDiffAE | 0.34 | 0.28 | 0.33 | 2.47±6.12 | 0.82 | 0.60 |
| raw | 0.97 | 0.97 | 0.97 | -2.17±2.10 | 0.99 | 0.79 |
| egDDIM | 0.88 | 0.85 | 0.43 | -0.82±3.03 | 0.98 | 0.67 |

We also perform the *de novo* random generation with randomly sampled $z \sim \mathcal{N}(0, I^d)$ (Figure 8, 9, 10). For each sampled $z$, we generate 10 molecules with random noise. Similar to previous tasks, the equivariant models outperform non-equivariant ones (Table 7). The general drop in

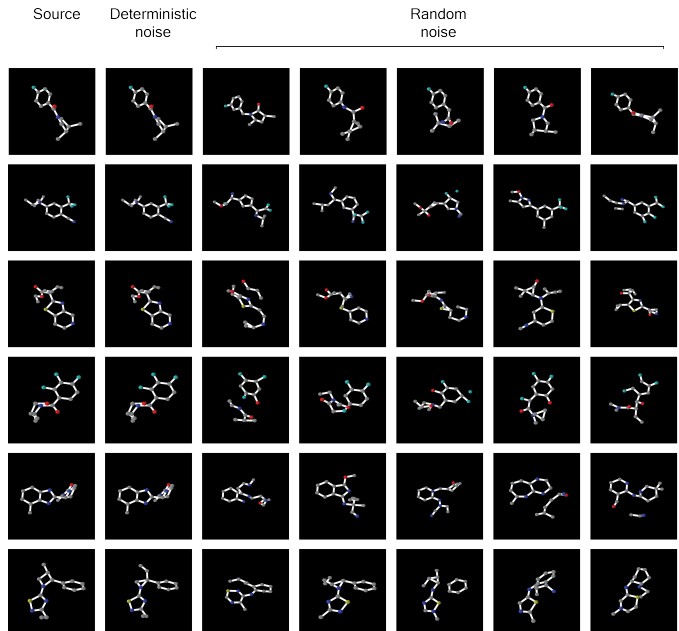

Figure 6: ZINC250k semantics-guided generation.

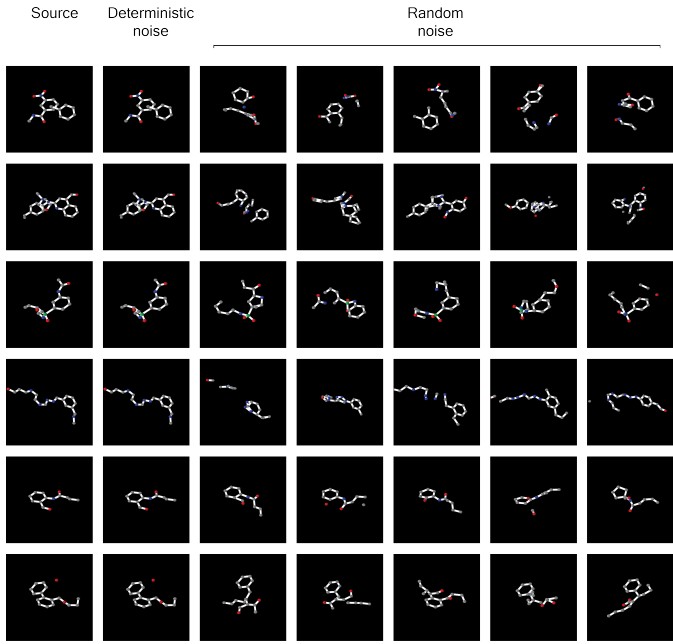

Figure 7: GEOM semantics-guided generation.

performance compared to semantics-guided generation, which only searches the proximity of valid molecules, indicates the existence of infeasible regions in the latent space that do not map to valid molecules. We believe the random generation performance could be further enhanced with more careful regularization and we leave it for future investigations.

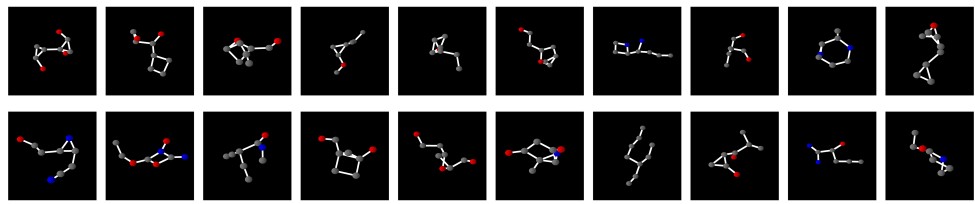

Figure 8: QM9 *de novo* generation.

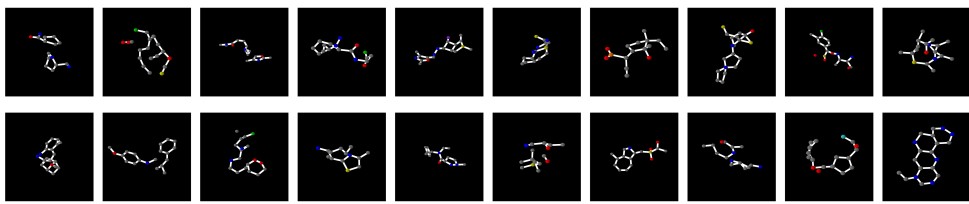

Figure 9: ZINC250k *de novo* generation.

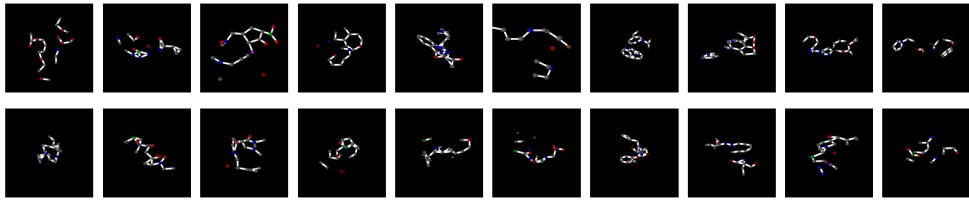

Figure 10: GEOM *de novo* generation.

## A.9 EXTENDED DETAILS AND RESULTS FOR PROPERTY MANIPULATION

Since labeled data may be limited in many application scenarios, we mask the property information during training and only use that from the test set. Specifically, we randomly select 1000, 1000, and 100 samples from the test set as the "PM-training/test/validation", respectively. We use the `egDiffAE` model trained on the original training set with no property information provided to obtain the semantics embeddings. We train the linear regression on the PM-training set. The property manipulation is then performed on the PM-test set.

We observe that the linear regression performs much better on $\alpha$ and $C_v$ than the rest (Figure 11), indicating the manipulation results for these two properties may be more reliable.

We normalize the property values for the training of the linear model and report the de-normalized value in the results. Detailed evaluation metrics are shown in Table 8. The metrics are averaged across all valid molecules.

## A.10 EXTENDED ABLATION STUDY

Table 9 shows the full ablation study results for property manipulation in 5.6.

Furthermore, we investigate the impact of node feature representation. Many previous studies use categorical features to represent atom types and directly apply a continuous diffusion model to them. This could work practically. However, as most dimensions of the one-hot encoding are not informative and depend on each other, such representation would be inefficient, especially with more types.

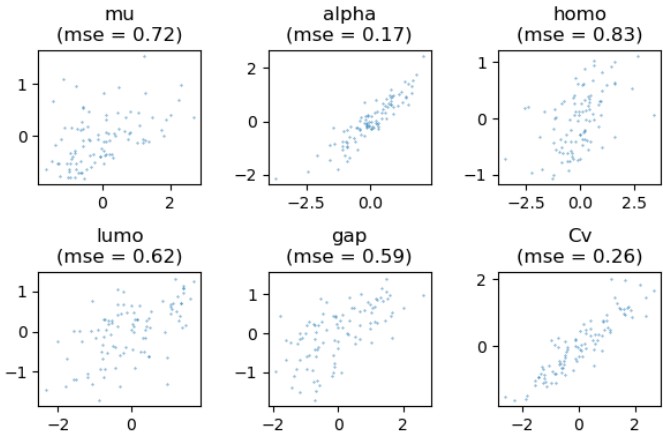

Figure 11: Linear regression on $z$ for property manipulation.

Table 8: Property manipulation results

| Model | $\mu$ | $\alpha$ | $\epsilon_{HOMO}$ | $\epsilon_{LUMO}$ | $\Delta\epsilon$ | $C_v$ |
|---|---|---|---|---|---|---|
| MAE↓ | | | | | | |
| egDiffAE | 1.67±1.1 | 4.88±4.72 | 0.02±0.01 | 0.04±0.04 | 0.05±0.04 | 2.99±2.44 |
| mpDiffAE | 1.50±1.01 | 10.7±11.47 | 0.04±0.03 | 0.10±0.03 | 0.12±0.04 | 2.60±2.03 |
| backprop | 0.58±1.05 | 0.75±3.04 | 0.03±0.03 | 0.02±0.03 | 0.01±0.02 | 3.65±4.88 |
| raw | 2.42±1.22 | 12.61±7.71 | 0.04±0.02 | 0.07±0.05 | 0.07±0.04 | 6.16±3.71 |
| egDiffAE-retrieval | 1.87±1.27 | 5.58±4.09 | 0.02±0.01 | 0.03±0.03 | 0.04±0.03 | 3.05±2.30 |
| similarity↑ | | | | | | |
| egDiffAE | 0.58±0.34 | 0.71±0.31 | 0.65±0.3 | 0.74±0.28 | 0.78±0.26 | 0.75±0.27 |
| mpDiffAE | 0.32±0.39 | 0.37±0.4 | 0.19±0.38 | 0.31±0.39 | 0.26±0.39 | 0.39±0.42 |
| backprop | 0.02±0.36 | 0.02±0.35 | 0.02±0.35 | 0.01±0.36 | 0.01±0.36 | 0.03±0.34 |
| raw | 1.00 | 1.00 | 1.00 | 1.00 | 1.0±0.0 | 1.00 |
| egDiffAE-retrieval | 0.58±0.35 | 0.73±0.30 | 0.6±0.34 | 0.73±0.27 | 0.74±0.27 | 0.75±0.29 |
| %valid↑ | | | | | | |
| egDiffAE | 0.52 | 0.86 | 0.65 | 0.84 | 0.83 | 0.75 |
| mpDiffAE | 0.24 | 0.37 | 0.00 | 0.31 | 0.29 | 0.36 |
| backprop | 0.00 | 0.00 | 0.00 | 0.00 | 0.00 | 0.00 |
| raw | 0.97 | 0.97 | 0.97 | 0.97 | 0.97 | 0.97 |
| egDiffAE-retrieval | 0.93 | 0.97 | 0.96 | 0.98 | 0.98 | 0.96 |
| energy↓ | | | | | | |
| egDiffAE | -2.17±2.84 | -0.19±2.96 | 2.79±4.3 | 1.09±3.87 | 0.11±3.59 | -2.08±2.69 |
| mpDiffAE | -1.72±2.86 | 3.10±4.00 | 4.06±0.97 | 4.19±6.01 | 3.55±3.52 | -0.50±3.29 |
| backprop | nan | -2.85±0.00 | nan | nan | nan | nan |
| raw | -2.41±2.11 | -2.42±2.11 | -2.41±2.11 | -2.41±2.11 | -2.41±2.11 | -2.41±2.11 |
| egDiffAE-retrieval | -3.13±1.91 | -1.48±1.81 | -0.80±1.87 | -1.51±1.82 | -1.92±1.78 | -2.79±1.89 |

Table 9: Ablation study.

| Model | $\mu$ | $\alpha$ | $\epsilon_{HOMO}$ | $\epsilon_{LUMO}$ | $\Delta\epsilon$ | $C_v$ |
|---|---|---|---|---|---|---|
| | Property manipulation %success ↑ | | | | | |
| base | 0.09 | 0.38 | 0.2 | 0.36 | 0.3 | 0.27 |
| prob- | 0.1 | 0.38 | 0.14 | 0.29 | 0.25 | 0.34 |
| MMD- | 0.08 | 0.4 | 0.15 | 0.35 | 0.3 | 0.33 |

Table 10: Comparison between bit and one hot representation.

| | Model | QM9 | | | |
|---|---|---|---|---|---|
| | | %valid | %unique | %novel | similarity |
| Semantics-guided | egDiffAE (bit) | 0.82 | 0.57 | 0.58 | 0.97±0.03 |
| | egDiffAE (onehot) | 0.56 | 0.48 | 0.49 | 0.95±0.04 |
| Random | egDiffAE (bit) | 0.66 | 0.45 | 0.62 | - |
| | egDiffAE (onehot) | 0.56 | 0.40 | 0.54 | - |

| Model | Property manipulation | | | | | |
|---|---|---|---|---|---|---|
| | %success ↑ | | | | | |
| | $\mu$ | $\alpha$ | $\epsilon_{HOMO}$ | $\epsilon_{LUMO}$ | $\Delta\epsilon$ | $C_v$ |
| egDiffAE (bit) | 0.12 | 0.48 | 0.30 | 0.43 | 0.34 | 0.32 |
| egDiffAE (onehot) | 0.02 | 0.27 | 0.12 | 0.21 | 0.17 | 0.16 |

On the other hand, existing categorical diffusion models, such as Hoogeboom et al. (2021), cannot be easily formulated into a DDIM for deterministic reverse sampling.

To overcome these limitations, we adopt analog bit representation (Chen et al., 2022), which is more dimension-efficient and can be directly modeled by continuous diffusion models. As in Table 10, the bit representation has an overall better performance than one-hot.

