# OpenReview forum: "3D Autoencoding Diffusion Model for Molecule Interpolation and Manipulation"
_ICLR.cc/2024/Conference — Submitted to ICLR 2024_

### Official Review · Reviewer_Xp9N · 2023-10-13

**Soundness:** 2 fair
**Presentation:** 2 fair
**Contribution:** 1 poor
**Rating:** 3
**Confidence:** 5

**Summary:**

This paper considers the problem of manipulating known molecules and interpolating between known molecules with deep generative models, especially diffusion models. Specifically, this paper focuses on the two tasks by the recently proposed equivariant diffusion generative models (requires equivariance since molecules are represented by 3D point clouds). The authors propose an auxiliary variable to improve the steerability and interpretability of the diffusion model. Experiments are conducted on both interpolation and optimization problems in the absence or the presence of a "template" molecule.

**Strengths:**

* This paper studies an important problem, how to interpolate between molecules and optimize molecules is a crucial problem in molecular discovery which has wide applications such as molecule optimization. The study of semantic directions in diffusion models is also timely and encouraged as it is the best-performing model in terms of generation quality at the moment.
* The idea of the method is quite simple but reasonable to introduce an auxiliary variable that learns control factors of the generative process.
* The proposed metrics seem to be reasonable for this relatively new task.

**Weaknesses:**

* The first major weakness of this paper is that it seems to ignore relevant literature in this area. Interpolation has been used and demonstrated for a long time [1, 2], the latent traversal has also been discovered before [3]. Despite this paper strikes the point that it focuses on diffusion models, but the authors should still consider proper comparisons or acknowledgments.

* Given this problem has been studied before, especially interpolation, the technical contribution of this paper is very limited. Another major contribution claimed in the paper (learning a semantics-guided autoencoding diffusion model) has also been proposed in [4]. The formulation looks exactly the same.

* It is not surprising that linear interpolation could lead to improved properties as studied widely in images and some in molecules (e.g. [3]), but the experimental details need to be included for a fair comparison and evaluation.

* The template-based manipulation seems an interesting setup, but a more realistic one should be specifying a specific molecule and then manipulate that molecule, given the stochastic forward process of diffusion model, maybe it is not possible. The name of "template-based" manipulation seems referring to manipulate any given molecule rather than a randomly sampled molecule.

* [Minor] The explanation of stability and validity seems to be different from the literature, see e.g. [5].

[1] Gómez-Bombarelli, R., Wei, J.N., Duvenaud, D., Hernández-Lobato, J.M., Sánchez-Lengeling, B., Sheberla, D., Aguilera-Iparraguirre, J., Hirzel, T.D., Adams, R.P. and Aspuru-Guzik, A., 2018. Automatic chemical design using a data-driven continuous representation of molecules. ACS central science, 4(2), pp.268-276.

[2] Zang, C. and Wang, F., 2020, August. Moflow: an invertible flow model for generating molecular graphs. In Proceedings of the 26th ACM SIGKDD international conference on knowledge discovery & data mining (pp. 617-626).

[3] Du, Y., Liu, X., Shah, N.M., Liu, S., Zhang, J. and Zhou, B., 2022. ChemSpacE: Interpretable and Interactive Chemical Space Exploration. Transactions on Machine Learning Research.

[4] Wang, Y., Schiff, Y., Gokaslan, A., Pan, W., Wang, F., De Sa, C. and Kuleshov, V., 2023. InfoDiffusion: Representation Learning Using Information Maximizing Diffusion Models. ICML 2023.

[5] Hoogeboom, E., Satorras, V.G., Vignac, C. and Welling, M., 2022, June. Equivariant diffusion for molecule generation in 3d. In International conference on machine learning (pp. 8867-8887). PMLR.

**Questions:**

See weaknesses.

---

> ### Author Response · Authors · 2023-11-23
>
> 1\. Comparison with interpolation and latent space optimization for 2D molecules
>
> Our focus is on semantics-guided diffusion in **3D molecular space**. We are aware of the existing works on interpolation and latent space optimization for molecules, but the majority of them are on 2D molecular graphs. The generation of 3D molecular structures poses different challenges such as preserving the symmetry, and incorporating semantic guidance into the generative process is a rarely explored topic. We will cite the mentioned papers, but it is not possible for direct comparison as these are different generation tasks.
>
> 2\. Contribution of the paper
>
> We have to disagree on this claim. Our main contribution is achieving semantic guidance on a **3D equivariant diffusion model**, which is a challenging task considering the complexity of the 3D space and the equivariance of the diffusion process. 3D interpolation is very different from that in 2D molecules or images. Also, learning informative semantics for 3D molecules is not trivial since we need a learning paradigm that can effectively leverage both the invariant and equivalent features. Please refer to the main rebuttal part 3 for extended discussion.
>
> 3\. Interpolation and experimental details
>
> As we have mentioned above, the generation of 3D molecules is a challenging task. Also, experimental details for the interpolation have been given in 5.4.
>
> 4\. Property manipulation
>
> We’d like to clarify that:
>
> (1) In fact, we are manipulating given molecule templates (which we now call “source” to avoid confusion). As described in 5.5, our property manipulation experiment takes the semantic embedding of a given source molecule, uses a simple linear model to alter its properties, and utilizes the altered embedding to generate a new molecule. We show in Fig 4 and Appendix Table 8 that molecules generated in this way largely resemble the source, while also gaining the altered property.
>
> (2) Our framework is based on DDIM, which allows for deterministic sampling. In other words, every embedding+noise combination could be deterministically mapped to a data point in the molecular space. Thus, the manipulated embedding also maps to a specific molecule.
>
> 5\. Stability and validity metrics
>
> We’d first like to clarify that our metric for “stability” is the same as Hoogeboom et al. (i.e. whether the allowed valency for the atoms is violated). However, as we are not modeling with explicit hydrogen, fragmented molecules can still be considered “stable” after adding the hydrogens back according to the allowed valency. This results in many false positives. Because of that, our “validity” metric adds another connectivity check, addressing the limitations of the previous metrics.
>
> Furthermore, for better comparison, we now report the validity metrics from MiDi, which are not very different from ours. Please see the main rebuttal part 4.1 for more details.

---

### Official Review · Reviewer_SYuH · 2023-10-26

**Soundness:** 3 good
**Presentation:** 2 fair
**Contribution:** 2 fair
**Rating:** 3
**Confidence:** 4

**Summary:**

The authors couple an autoencoder with a generative diffusion-based model for 3D molecule generation to overcome the potential limitation of not being able to operate on latent embeddings when using diffusion models. A semantic latent is trained as a condition for guiding the diffusion model to manipulate / interpolate molecules in 3D space. The authors present three different scenarios to show the effectiveness of the proposed method. One of the scenarios measures the ability to manipulate the latent embedding towards a target property using a linear regression model, which could be used, e.g., for the rational design of molecules.

**Strengths:**

- Although many works have been proposed in recent years leveraging the latent embeddings of AE/VAE architectures for interpolation and optimization tasks, the combination of a diffusion model with an autoencoder provides some novelty.

**Weaknesses:**

- Tables and figures are insufficiently described in the captions.
- The paper only presents validity and stability metrics, but no similarity measures to examine the effectiveness of the template-guidance. Beyond that, the quality of the 3d geometries are not evaluated (e.g. energies and atomic forces).
- egDiffAE does not perform well in terms of stability (e.g. QM9 template-based generation 85.8% vs de-novo methods like EDM  90.7% [Hoogeboom et al] / Midi 97.5 [Vignac et al 23]). GEOM mol stability metrics are not given (only atom stability).
- The property manipulation does not seem to meaningfully alter the template, but mostly move atoms around and break/form some bonds. As a result, the results of property manipulation do not appear to be in equilibrium, but arbitrary conformations. Energies and atomic forces need to be carefully checked, since the optimized molecular properties are only meaningful in equilibrium. Beyond that, as far as I can tell, the property models have been trained on equilibirum molecules, i.e. they might be out of their domain of applicability at arbitrary conformations.
- In Table 3, "retrieval" baseline yields similar results to egDiffAE, in some properties even performing better. For the the remaining properties, it is not possible to determine whether the improvement is significant and meaningful without error bars and units.

**Questions:**

- What is the practical relevance of molecule interpolation?
- How close are the generated / manipulated molecules to equilibrium?
- Why use MMD regularization instead of KL prior regularization as in VAE?
- How large is the batch size? MMD usually requires large batch sizes
- From the figures, it seems that interpolation and manipulation is only possible for molecules with same number of atoms?
- Why are there are no figures showing results on GEOM? This might be important to evaluate how the generation performs on larger molecules?

---

> ### Author Response · Authors · 2023-11-23
>
> 1\. Captions
>
> Thanks for the comment. We have already discussed all tables and figures extensively in the Results section. We also add some descriptions to the captions in the revision.
>
> 2\. Similarity and energy metrics
>
> Thanks for the comment. We now also report similarity measures and energy estimated by xTB. Please find more details in the main rebuttal part 4.
>
> 3\. Comparison with *de novo* generation
>
> We’d like to point out that the template-based (which we now call “semantics-guided” to avoid confusion) generation in our study is not directly comparable with *de novo* generation results in those works because:
>
> (1) Our generation task is different from *de novo*. Please refer to the main rebuttal part 2 for more details.
>
> (2) We do not explicitly include hydrogen. As mentioned in the main rebuttal part 4.1, the implicit hydrogen also leads to different validity metrics as in those works.
>
> We’ve also shown *de novo* generation results in Appendix Table 7. We believe there is room for improvement in the quality of both tasks in future studies.
>
> 4\. Metrics for GEOM
>
> Our model is based on the framework of EDM, which only models the atom types and coordinates, and tends to produce disconnected structures for large molecules (Appendix A8). Thus, only atom stability is reported for GEOM, as in EDM.
>
> 5\. Property manipulation
>
> First of all, we have examples where atom types are modified in Fig 4. We also have discussed the limitations of the current results in Conclusions.
>
> As our model is trained on QM9, which only contains equilibrium molecules, we’d assume the generated molecules should resemble equilibrium ones. Thus, property models on equilibrium molecules could be applied to the generations, similar to EDM.
>
> We use xTB to estimate the energy of the molecules. As shown in Appendix Table 8, for most properties, the energy distribution of the generated molecules shifts slightly higher from the test set (equilibrium), which means most of the generated molecules have near-equilibrium conformations.
>
> 6\. Retrieval
>
> We’d like to clarify that the “retrieval” method uses the **semantic embedding learned from egDiffAE** to directly select **molecules from the training set** with the desired property. This has been explained in 5.5. It is another way of obtaining molecules with desired compositions and properties using our semantics embedding. Its high performance is an expected result because:
>
> (1) It searches for real molecules, so validity is guaranteed.
>
> (2) It shows our semantic embedding contains sufficient information for the property-determining components, so if a known molecule has a similar embedding as z’, it means that the molecule should carry the same property. This further proves the effectiveness of the semantics embedding.
>
> On the other hand, this baseline does not meaningfully generate any new molecules but only searches among the training set. Thus, it mostly offers a sanity check and an approximate upper bound of the performance. We have further clarified that in Table 3 and Section 5.5.

---

> > ### Author Response · Authors · 2023-11-23
> >
> > (continued)
> >
> > 7\. Relevance of interpolation
> >
> > With interpolation, we could effectively explore and exploit the molecular space looking for novel “intermediate” molecules with properties of interest, which is an extremely challenging problem due to the huge space for 3D molecules.  For example, based on two different molecules with similar properties, we can find more novel candidates with similar properties and “intermediate” structures if we can achieve a smooth interpolation.
> >
> > 8\. Equilibrium of the generated/manipulated molecules
> >
> > As mentioned above, we use xTB to evaluate the energy of the generated/manipulated molecules (Table 1, Appendix Table 8). We find most of the generated molecules have near-equilibrium conformations from the energy distribution.
> >
> > 9\. MMD and KL divergence
> >
> > Theoretically, KL regularization could achieve similar results, but it is known to be unstable to train and subject to problems like posterior collapse [1], and requires careful tuning.
> >
> > [1] Bowman, Samuel R., et al. "Generating sentences from a continuous space." arXiv preprint arXiv:1511.06349 (2015).
> >
> > 10\. Batch size
> >
> > The batch size is 128 molecules, as stated in Appendix A.5.
> >
> > 11\. Number of atoms
> >
> > As the diffusion model requires an initial noise with a specified atom number, it is more straightforward to use the same atom number, especially for interpolation where we also need to interpolate the initial noises. Nevertheless, we could also potentially work with different atom numbers, but it adds more to the complexity of the evaluation.
> >
> > 12\. Figures for GEOM
> >
> > Thanks for the suggestion. We add GEOM generation results in Appendix Fig 7 and 10. Similar to ZINC, the generations on GEOM tend to be disconnected. This is a known limitation with models solely relying on coordinates of atoms like in EDM, as is discussed in Appendix A8. For future work, we could improve the generation quality on larger molecules by explicitly modeling the bonds, etc.

---

### Official Review · Reviewer_bMs7 · 2023-10-30

**Soundness:** 2 fair
**Presentation:** 2 fair
**Contribution:** 2 fair
**Rating:** 3
**Confidence:** 3

**Summary:**

In this paper, the authors propose an augmentation to the 3D equivariant diffusion model proposed by Hoogeboom et al, 22. This is done by including an extra “semantic embedding” (from an auxiliary encoder) at each step of the reverse diffusion process. The authors show that the “semantic embedding” improve some generation metrics  (although the choice of metrics are not the best, see below) and achieve good “property manipulation” performance. Experiments are done on three standard molecule datasets (QM9, GEOM-drugs and ZINC

**Strengths:**

- As mentioned in the paper, manipulate the latent space of diffusion models is not as easy as in other generative models. Understanding and improving on this issue is valuable for the research community.
- Moreover, the problem tackled on this work (ie, 3D molecule generation) is an important (and under-explored) application for machine learning.

**Weaknesses:**

- The paper is not very well written. Many definitions are wrongly used or used too loosely (eg, “de novo” or “template-based”) and a lot of implementation details are missing, making reproduction difficult.
- There are citations missing (many work do some kind of semantic guiding, eg Hoogeboom et al, 22, all the work on pocket-conditioning, linking, fragment-based generation, etc).
- The idea of conditional generation in some semantic space is interesting. However, the way it has been proposed seems ad-hoc to me. It is unclear what the “semantic encoding” is learning if the only loss is MMD with Gaussian.
- The metrics used on the experiments are not very convincing. The authors only use validity and atom stability—metrics that are known to be not very informative of the quality of the samples. I would recommend the authors to use the MiDi metrics (Vignac et al, 23) at a minimum. It is also not clear to me that the “smootheness” metric used is very informative. The results shown on Table 2 (mean +- std for both “smootheness mean” and “smootheness std” are not clear).
- The authors do not compute any measure of uniqueness/diversity of the generated molecules. It would be nice to see how diverse are the generated samples in this setting (since it is very important to generate diverse molecules in practice). I would imagine that the “semantic embedding” could highly reduce the diversity and this could be bad for some applications.

**Questions:**

- Please see the weaknesses above. It would be nice to hear the authors’ opinion about them.
- Many work propose some kind of inpainting for scaffold/linking (eg, DiffHopp (Torge et al 23), SBDDDiff (Schneuing et al 22), DiffLinker (Igashov et al 22), etc). It would be good to compare the performance of the model with some inpainting baseline. How does the proposed method differs from those approaches?

---

> ### Author Response · Authors · 2023-11-23
>
> 1\. Use of terms and experimental details
>
> We believe *de novo* is properly used as we are randomly sampling molecules from the molecular space without any specification in the respective experiments. We do acknowledge that “template-based” may be misleading so we changed that wording to “semantics-guided”, reflecting the real nature of our experiment. Please see the main rebuttal part 1 for more details.
>
> Also, we’ve included many implementation details in the Appendix, including on the data processing (A.3), model architecture (A.4), and model hyperparameters and training (A.5). We will make our references in the main text to the Appendix clearer. Also, if you could specify the missed implementation details, we will be happy to clarify and add them.
>
> 2\. Comparison with ligand and linker designs, and inpainting (also responding to question 2)
>
> It is noteworthy that those works do not have the same “guidance” as ours. We consider those works as more of an externally **conditioned** generation, where a molecule (or a part of it) is generated conditioned on known pockets/fragments. There is no explicit control over the “semantics” (i.e. the composition and shape) of the generated molecule itself in those problems. Here our semantics contains direct information about the entire molecule, rather than the condition. Please refer to the main rebuttal part 2. We’ve also discussed those works in the Related Works.
>
> We do believe these are interesting applications for 3D diffusion models, and that our framework could be applied to them by leveraging both the condition and the explicit control of certain parts of the structure, and compared with those papers. However, it may require major modifications to the current training and generation framework to perform inpainting, so we’d like to leave it for further studies.
>
> 3\. The loss for the semantics embedding
>
> We’d like to clarify that the semantic embedding is used as an input to guide the diffusion process. Thus, the diffusion model’s noise estimation loss (equivalent to the reconstruction loss in traditional autoencoder settings) is also back-propagate to the encoder, and the MMD loss is an additional regularization term. Thus, as is proved in 4.3, it also contains information about the input molecule. This is similar to an autoencoder framework where the diffusion model effectively functions as a “decoder”.
>
> 4\. Evaluation metrics
>
> 4.1 Validity
>
> Thanks for the suggestion. We now report the same validity metric as in MiDi. Interestingly, both MiDi’s valid metric and our previous valid are both stability (valency not violated) plus connectivity (successfully sanitized by RDKit), the results are almost the same as before.
>
> 4.2. Atom stability
>
> For larger molecules, because we are building upon EDM’s framework (see main rebuttal part 2), we encounter the same problem where the generations are fragmented. Thus, we also rely on only atom similarity like in EDM. We could improve the generation quality on those datasets by incorporating some more recent advances in the 3D modeling of molecules (e.g. explicitly generating the bonds). We’d leave it for future studies.
>
> 4.3 Interpolation
>
> We have explained in detail the smoothness metrics in 5.4. Basically, for an interpolation trajectory, our desired characteristics are:
>
> (1) Overall smooth transitions between neighboring points (low smoothness mean)
>
> (2) Near-uniform steps taken across the trajectory (low smoothness std)
>
> (3) The midpoint being a proper mixture of the two endpoints, rather than something entirely different (high midpoint similarity)
>
> Specifically for (3), we can observe in the interpolation with standard DDIM that the midpoint is a new molecule that does not resemble either of the endpoints. Thus, it is not a smooth trajectory from one end to the other.
>
> 5\. Novelty and uniqueness
>
> Thanks for the comment. We now report the uniqueness and novelty using the same metrics as in MiDi (see Table 1 and Appendix A.7.3).
>
> As shown in Table 1, we expect molecules generated with the same semantics embedding to have lower diversity than de novo generations. This is the exact purpose of our framework: to enforce sufficient control on the composition, shape, etc. of the molecule so that the same semantic embedding will result in mostly similar molecules regardless of the random noise. Then we can explore the molecular space with finer control over the direction.
>
> If the goal is to generate diverse molecules, we could randomly sample a semantic embedding and use it for generation, as in Appendix Table 7.

---

### Official Review · Reviewer_6Nzc · 2023-11-01

**Soundness:** 3 good
**Presentation:** 1 poor
**Contribution:** 2 fair
**Rating:** 5
**Confidence:** 2

**Summary:**

This submission proposes a method to control the generation process of 3D molecule diffusion models. This is to mitigate the research gap that most existing works focus on de novo molecule design, aka, generate random molecules, without controlling the molecule's properties. Specifically, they utilize an auxiliary encoder's output to condition the generation process of an equivariant diffusion model.  This allows the direct operation on the embeddings for effective interpolation and property manipulation of template molecules.

**Strengths:**

* The studied task of controlled molecule generation is of significant research and application value. Compared to de novo moleucle generation, controlled moelcule generation aligns better with the practical usage.

**Weaknesses:**

I am not very familar to diffusion models and the generation of 3D molecules. To me, the biggest concern is on the evaluation of the proposed method. It seems that there are no baselines that are directly comparable for the main experiments (Table 1, Table 2, Table 3), and the used evaluation metrics are sometimes insufficient to prove the value of the proposed method. For example:

* In Table 1, only `stableness` and `validity` are used as evaluation metric. However, for a template-based generation task, I expect metrics like `similarity to the template`, and `diversity of the generated molecules`.
* In Table 2, the used evaluation metrics are somehow confusing. For `smoothness` and `midpoint similarity`, how are the similarity scores calculated? Do you use cosine similarity on embeddings? Or, do you apply other similarity metrics direclty on molecules?



I am willing to re-evaluate my rating based on other reviewers' comments.

**Questions:**

n/a

---

> ### Author Response · Authors · 2023-11-23
>
> We acknowledge that there aren’t many existing baselines to compare with our method, except for interpolation where we could directly compare with a simple DDIM. For example, regarding molecular properties, most existing studies rely on conditioned generation given the properties, with little to no control over other aspects of the generated molecule. In this study, we seek finer control on not only the properties but also the composition, overall shape, etc. with the semantics embedding. Thus, for most parts of our work, we are trying to prove the generated molecules are of good quality and could achieve the conditioning we want.
>
> As mentioned in the main rebuttal part 3, here we try to demonstrate the usefulness of semantics guidance for the generation on a very well-established basis and dataset. We believe this framework could be extended to achieve finer guidance on more complex generative tasks, such as linkers and pocket-based ligands, by incorporating some recent progress in 3D molecule modeling dedicated to those tasks.
>
> 1\. Evaluation metrics
>
> Thanks for the suggestion. We’ve included similarity and uniqueness (i.e. diversity) in the results. Please refer to the main rebuttal part 4.3 for more details.
>
> 2\. Interpolation metrics
>
> The similarity scores are defined as the cosine similarity between the embeddings. For an interpolation trajectory, our desired characteristics are:
>
> (1) Overall smooth transitions between neighboring points (low smoothness mean)
>
> (2) Near-uniform steps taken across the trajectory (low smoothness std)
>
> (3) The midpoint being a proper mixture of the two endpoints, rather than something entirely different (high midpoint similarity)
>
> Specifically for (3), we can observe in the interpolation with standard DDIM that the midpoint is a new molecule that does not resemble either of the endpoints. Thus, it is not a smooth trajectory from one end to the other. Please find more detailed descriptions in 5.2 and 5.4.

---

### Author Response · Authors · 2023-11-23
**Main Rebuttal**

We thank all reviewers for their comments. Here we’d like to address some common concerns raised by several reviewers. Specifically, we believe some important points were not properly delivered, and need to be clarified here and further in the updated manuscript (major changes are highlighted in blue).

1\. Clarifications of “template-based”

We acknowledge that this term is somewhat misleading. In this work, we aim to generate new molecules with similar composition and structure (the “semantics”) as a known one, but potentially different properties. The same applies to property manipulation. We’re not adding functional groups to molecular scaffolds, but that could also be done within our framework with a re-designed diffusion process. To avoid confusion, we now call the generation process “semantics-guided” because it uses the embedding to generate semantically similar molecules. We have accordingly revised our paper.

2\. Difference from existing works

We’d like to clarify that our generation task is different from *de novo* methods. In our case, we are generating 3D molecules that are semantically similar to a known one (the source), by controlling its composition, shape, and multiple properties. This is a challenging and less explored task with little benchmarking, and cannot be done with *de novo* generation where there is no guidance.

Our generation task also differs from existing conditioned generation methods, such as the linker designs and pocket-based ligand designs. In those cases, the generation task is more like inpainting: completing a 3D structure by conditioning on part of it. The semantics we use, however, contains information about a complete molecule. It could be extracted and modified from given molecules to explore the space around them, as in our experiment. Of course, we could also achieve guided generation for the aforementioned tasks, e.g. generating a ligand given the binding pocket (external conditioning) AND with control over its compositions, 3D structures, and properties (the semantics). We’d leave that for future work.

3\. The main contribution of our paper

(1) The purpose of our work is to show the usefulness of semantic guidance for 3D molecular generation. Thus, our model is based on a well-established 3D diffusion framework (EDM) [1] and a widely-used benchmark dataset (QM9). EDM does have its limitations, such as failure in large molecules, which we also observe in our study. We could in principle scale up the framework for larger molecules and more complex tasks (see Conclusion).

(2) From another perspective, we could obtain information-rich representations for 3D molecules. It is challenging to handle equivariance in a lower-dimensional space. An equivariant diffusion-based framework can easily separate the invariant features (controlled by the semantics embeddings in our case) from the equivariant features (preserved during generation). Also, as proved in 4.3, the embeddings share high mutual information with the molecules, making them highly informative. For example, in the “retrieval” method in Table 3 (5.5), we could use our embeddings to query molecules with both desired compositions and properties.

4\. Evaluation metrics

4.1 Validity

Unlike some existing works, we are not including hydrogens, as they do not carry much information and may complicate the training. During evaluation, they will be added back according to valency constraints. This will bloat the common stability measure. Thus, we also consider connectivity, resulting in a similar validity metric as in MiDi [2]. For better clarity, we now report MiDi’s metrics, and the results are almost the same.

4.2 Similarity

We agree that similarity metrics are needed. Comparing 3D structures is a challenging task as we cannot directly compare their 2D molecular graphs. Shown in [3], there are cases where molecules are similar in 2D but not in 3D, and vice versa. For this purpose, we train a separate egDiffAE model and use its embedding for the similarity evaluation.

4.3 Uniqueness and novelty

We now also report the uniqueness and novelty measure from MiDi [2], based on the canonical SMILES strings. These metrics are not meaningful for ZINC and GEOM because the bond length-based reconstruction (as in [1]) does not work well on large molecules, resulting in nonsense SMILES.

4.4 Energy

We use xTB to estimate the energy of the generated molecules.

Please find detailed explanations to 5.2 and Appendix A.7. Table 1 and 3 are updated accordingly.

[1] Hoogeboom, Emiel, et al. "Equivariant diffusion for molecule generation in 3d." International conference on machine learning. PMLR, 2022.
[2] Vignac, Clement, et al. "Midi: Mixed graph and 3d denoising diffusion for molecule generation." arXiv preprint arXiv:2302.09048 (2023).
[3] Axen, Seth D., et al. "A simple representation of three-dimensional molecular structure." Journal of medicinal chemistry 60.17 (2017): 7393-7409.

---

### Meta-Review · Area_Chair_MHNL · 2023-12-08

**Metareview:**

The paper presents a semantic-guided equivariant diffusion model for controllable molecular generation. While the paper presents an interesting direction to controllable generation, the reviewers criticized the submission for lack of related baselines and quantitative metrics, presentation quality, as well as missing discussions on prior works. The rebuttal attempted to address some of these issues with major changes to the original submission. However, the reviewers and AC do not find the paper ready for acceptance at ICLR.

**Justification For Why Not Higher Score:**

The reviewers unanimously rated the paper below the acceptance threshold

**Justification For Why Not Lower Score:**

N/A

---

### Decision · Program_Chairs · 2024-01-16

Reject